# Conditional Generative Quantile Networks via Optimal Transport

**Jesse Sun[1], Dihong Jiang[1,2], Yaoliang Yu[1,2]**
University of Waterloo[1], Vector Institute[2]
{j294sun, dihong.jiang, yaoliang.yu}@uwaterloo.ca

## Abstract

Quantile regression has a natural extension to generative modelling by leveraging a stronger pointwise convergence than in distribution. While the pinball quantile loss works well in the scalar case, it cannot be readily extended to the vector case. In this work, we propose a multivariate quantile approach for generative modelling using optimal transport with provable guarantees. Specifically, we suggest that by optimizing smooth functions parameterized by neural networks with respect to the dual of the correlation maximization problem, the function uniformly converges to the optimal convex potential. Thus, we construct a Brenier map as our generative quantile network. Furthermore, we introduce conditioning by approximating the convex potential using a first-order approximation with respect to the covariates. Through extensive experiments on synthetic and real datasets for conditional generative and probabilistic time-series forecasting tasks, we demonstrate the efficacy and versatility of our theoretically motivated model as a distribution estimator and probabilistic forecaster.

## 1 Introduction

A rudimentary result in probability theory states that passing a uniformly random sample from the unit interval through the univariate quantile function results in a random sample from a desired cumulative distribution function (CDF). Quantile functions have been widely explored in real applications, e.g. measuring financial risks (Rüschendorf, 2013), building regression functions at various levels (Koenker & Gilbert, 1978), robust statistical estimators under model mis-specification (Koenker, 2005). The pin-ball loss in quantile regression has also been used in training supervised classifiers (Huang et al., 2014) and building unsupervised generative models (Ostrovski et al., 2018).

The quantile function estimator is a generative model when it learns the entire target distribution by varying different quantile levels. There are a few major challenges for quantile regression: (1) While traditional quantile regressors tend to be trained at fixed quantile levels, the entire quantile function normally obtained through interpolation or smoothing suffers from the quantile crossing problem, i.e. the monotonicity property of the quantile function is violated. (2) Although the pin-ball loss works well for the univariate case, its extension to multivariate setting, i.e. a setting of paramount practical importance and is our focus in this work, remains a challenge. While in principle one can apply quantile regression to each dimension, as in Ostrovski et al. (2018), this naive approach ignores the high-order correlations between components and can lead to incorrectly learned models.

Interestingly, Carlier et al. (2016) suggests that multivariate quantile functions can be characterized as solutions to an optimal transport problem under mild assumptions, via the gradients of a convex function (Brenier, 1991; McCann, 1995). Note that gradients of convex functions are monotone operators, and hence coincide with the non-crossing property of quantile functions.

In this work, we seek to generalize univariate generative quantile modelling to the multivariate setting by extending the vector quantile regression of Carlier et al. (2016; 2017) that constitute building blocks of our work. Our primary focus is on multivariate time-series data forecasting, i.e. treating prediction at a time point conditioned on history data as a probability distribution rather than an exact value. A probabilistic approach to time-series forecasting is of great prominence as risk and confidence levels are automatically captured leading to better-informed decisions by decision-makers that analyze and act on these predictions. In addition to time-series forecasting,

we also test our generality on image generation tasks, i.e. treating the learned multivariate quantile function as a map from uniform distribution to the target distribution. Our contributions can be summarized as follows: (1) We adopt the dual formulation of correlation maximization in Carlier et al. (2016; 2017), and extend it to conditional generative modelling *in both time-series forecasting and image generation* to demonstrate its effectiveness and versatility. (2) In contrast to the linear parameterization in Carlier et al. (2016; 2017), we propose to significantly increase model capacity by parameterizing the model as input convex neural networks, which can now be optimized using large-scale optimization algorithms and allow for broader applicability across a wider range of tasks. (3) While Carlier et al. (2016; 2017) condition on raw covariates, we propose to condition on features of covariates encoded by LSTM layers, which is more scalable and can be jointly trained. We defer some background material on quantile networks and optimal transport to Appendix A.

## 2 Method: Brenier Maps as Generative Quantile Networks

We propose a novel method for multivariate density estimation using a quantile approach that allows for flexible task-dependent conditioning. To extend to conditioning, a first-order approximation of the convex potential is made with respect to the covariate embeddings.

### 2.1 Multivariate Quantiles as Brenier Maps

Suppose $\nu$ and $\mu$ are two distributions. Let $Q_\nu(u) : [0,1] \to \mathbb{R}$ be defined as a univariate quantile function of $Y \sim \nu$. By definition, quantile functions are monotonically increasing so $Q_\nu(u)$ will be maximally correlated with $u$. A quantile function is thus a solution to the correlation maximization problem (Galichon & Henry, 2012):

$$\int_0^1 u Q_\nu(u) \mathrm{d}u = \max_{U \sim \mathrm{Unif}([0,1])} \mathbb{E}_Y(UY). \tag{1}$$

Now, we are interested in generalizing this equality to the multivariate setting for $U, Y \in \mathbb{R}^d$ where $U \sim \mathrm{Unif}([0,1]^d)$. For the remainder of this paper, $\mu$ and $\nu$ are $d$-dimensional distributions.

Let $L_d^p$ denote the $L^p$ space with range space $\mathbb{R}^d$.

**Definition 1** (Maximal Correlation Functionals, Galichon & Henry 2012). *A functional $\rho_\mu : L_d^2 \to \mathbb{R}$ is called a maximal correlation functional with respect to distribution $\mu$ if for all $W \in L_d^2$,*

$$\rho_\mu(W) := \sup \left\{ \mathbb{E}_U[U^\top W], U \sim \mu \right\}. \tag{2}$$

The distribution $\mu$ should be absolutely continuous with respect to the Lebesgue measure, then there exists a closed convex semi-continuous function $g : \mathbb{R}^d \to \mathbb{R}$ such that $Y = \nabla g(U)$ $\mu-$almost surely (Villani, 2008), which motivates us to establish a multivariate extension of (1) using the gradient of convex $g$ that is maximally monotonic. Furthermore, $\nabla g$ achieves an optimal coupling between $\mu$ and $\nu$ (Huang et al., 2021) such that $\rho_\mu(X) = \left\{ \mathbb{E}[U^\top \nabla g(U)], U \sim \mu \right\}$. By Brenier's theorem:

**Theorem 1** (Brenier 1991). *If $Y$ is a squared-integrable random vector in $\mathbb{R}^d$, there is a unique map of the form $T = \nabla g$ for some convex function $g$ such that $\nabla g_\# \mu = \nu$.*

We call $\nabla g$ a Brenier mapping between $\mu$ and $\nu$. Now, we are ready to define:

**Definition 2** (Convex Potential Quantile (CPQ), Chernozhukov et al. 2017; Hallin et al. 2021). *Let $g : \mathbb{R}^d \to \mathbb{R}$ be a closed convex function. Assume $\mu$ is absolutely continuous on $\mathbb{R}^d$ with respect to the Lebesgue measure. Then, the Convex Potential Quantile of $Y \sim \nu$ is defined as $Y = \nabla g(U)$ where $U \sim \mu := \mathrm{Unif}([0,1]^d)$, i.e. $\nabla g_\# \mu = \nu$.*

We defer the estimation of CPQ to Appendix B.

### 2.2 Conditional Generative Quantile Network

Modelling time-series trajectories can be viewed as a conditional generaive task, since the outcome often depends on the history and other covariates. Let $X \in \mathbb{R}^m$ be a random vector to be conditioned

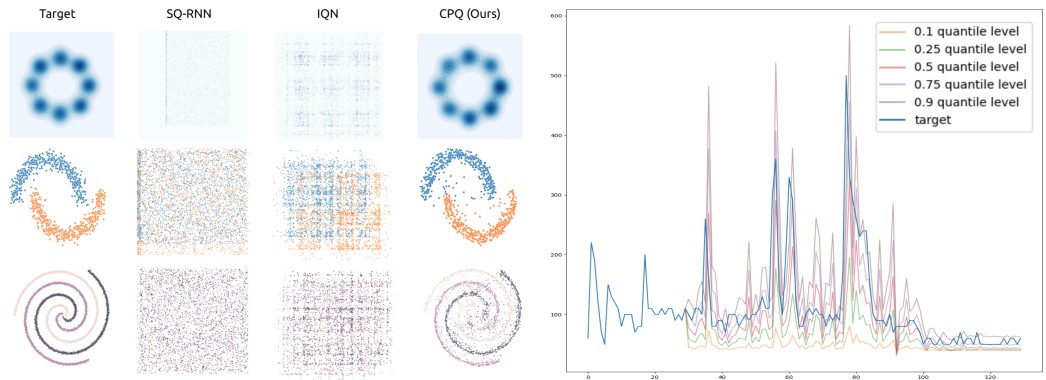

Figure 1: **Qualitative results on 2D density estimation (left) and estimated quantile levels of the trajectory on univariate time-series (right). Left:** The first column shows target distributions, while the following three columns show the generation by three different methods. It can be observed that our proposed method (CPQ) is able to capture the inter-dimensional correlation, whereas the other two baselines cannot. **Right:** the distribution of the trajectory is inferred which can be used to measure the uncertainty and risk of the trajectory prediction.

on, now we are interested in generating samples $Y$ given $X$ and learning the conditional convex potential $\varphi_X(U)$. Fixing $U$, we take the first-order approximation about $X$,

$$\varphi_X(U) = \varphi(U) + b(U)^\top X. \tag{3}$$

where $\varphi(U) : \mathbb{R}^d \to \mathbb{R}$ and $b(U) : \mathbb{R}^d \to \mathbb{R}^n$ are smooth functions. However, Carlier et al. (2016) only set $\varphi$ and $b$ as linear functions. To allow for greater expressivity in less trivial tasks such as time-series forecasting, we propose $\varphi$ and $b$ could be parameterized as input convex neural networks or smooth neural networks. We direct the reader to Appendix E.4 for more details on implementation. While our assumption of $X$ as a random vector can certainly hold for time-series forecasting by constructing a Markov model such that $X_{t+1} := \nabla\varphi_{X_t}(U)$, going beyond the Markov assumption, we propose a model that capitalizes on the advances of deep sequence models,

$$\varphi_X(U) = \varphi(U) + b(U)^\top f(X). \tag{4}$$

Under this formulation, $f$ is any task-specific mapping to yield latent embedding vector $f(X)$. In our application, $X_t := [x_{t-h+1}, \ldots, x_t] \in \mathbb{R}^{m \times h}$ is a sequence of vectors where $h$ is the length of our history, and $f$ is any deep auto-regressive sequence model. By learning a non-linear transformation of $X$, high-order moments between components of $X$ can be modelled, thereby leading to a more expressive conditional generative quantile model.

With the parameterization in Eq. (4) and introducing mini-batches, problem (12) now becomes:

$$\min_{(\varphi, b)} \sum_{i=1}^{N} \varphi(U_i) + \max_{j \in [N]} U_j^\top Y_i - \varphi(U_j) - b(U_j)^\top f(X_i), \tag{5}$$

subject to zero-mean decorrelation constraints in Eq. (13). The constraint can be enforced by applying batch normalization to the $f(X)$ vector without the translation term such that the empirical batch mean of $f(X)$ is zero-centered. We also find that enforcing this constraint is necessary for numerical stability and theoretical guarantees. The problem now becomes unconstrained, so gradient descent optimizers can be directly used. For theoretical analysis of our model specification and estimation, we refer the reader to Appendix C.

## 3 EXPERIMENTS

In this section, we evaluate our proposed approach to conditional generative modelling on a variety of experiments ranging from (conditional) distribution estimation of synthetic 2D data to probabilistic time-series forecasting on *UCI Appliances Energy* and *Google Stocks* data. Full details on implementation details can be found in Appendix E.4. For evaluation, we compute the max and

Table 1: Performance evaluation on the multivariate time-series datasets `Energy` and `Stocks`. Results are averaged over 5 runs. Lower score is better. We use boldface for the lowest score.

| Dataset | Model | MaxAE | MeanAE | QL50 | QL90 | RMSE | sMAPE |
|---------|-------|-------|--------|------|------|------|-------|
| Energy | DeepAR | 1.033 | **0.049** | — | — | 0.103 | 0.203 |
| | TimeGAN | 0.970 | 0.129 | — | — | 0.172 | 0.378 |
| | TempFlow | 1.236 | 0.052 | — | — | 0.110 | **0.198** |
| | IQN | 0.980 | 0.050 | **0.025** | **0.015** | 0.094 | 0.212 |
| | CPQ (Ours) | **0.624** | 0.051 | 0.026 | 0.038 | **0.091** | 0.210 |
| Stocks | DeepAR | 0.791 | 0.067 | — | — | 0.092 | 0.408 |
| | TimeGAN | 1.379 | 0.092 | — | — | 0.181 | 0.256 |
| | TempFlow | 0.670 | 0.040 | — | — | 0.063 | **0.213** |
| | IQN | 0.740 | 0.026 | 0.013 | **0.007** | 0.041 | 0.220 |
| | CPQ (Ours) | **0.660** | **0.024** | **0.012** | 0.014 | **0.036** | 0.220 |

mean absolute errors (MaxAE and MeanAE), the $50^{th}$ and $90^{th}$ quantile loss (QL50 and QL90), the root mean squared error (RMSE), and the symmetric mean absolute percentage error (sMAPE). Descriptions of competitor algorithms are given in Appendix E.2.

### 3.1 SYNTHETIC 2D EXPERIMENT

To demonstrate the shortcoming of previous quantile methods in learning joint distributions, we test our method against IQN and Spline Quantiles in modelling highly-correlated 2D distributions. The experimental details are deferred to Appendix E.5. Indeed, this experiment provides evidence for our suspicion that prior quantile methods cannot readily be extended to the multivariate case, i.e. minimizing the sum of marginals of these respective losses is insufficient to modelling higher-order dynamics between variates.

### 3.2 PROBABILISTIC TIME-SERIES FORECASTING

We consider both univariate and multivariate cases. The univariate experiment is given in Appendix F.1. We evaluate our method on multivariate probabilistic forecasting using the full *UCI Appliances Energy* dataset and *Google Stocks* data. The experimental details are given in Appendix E.6. Comparisons are summarized in Table 1, where our CPQ yields comparable or superior performance and indicates more robustness across datasets and metrics than baselines.

The advantage of a quantile approach to distribution estimation as the exact point-wise quantile level can be extracted in a single forward-pass by exploiting a monotonic parameterization of the density estimator which cannot be done so using traditional generative models. Another hidden advantage of our quantile approach in the multivariate setting is that individual quantile levels can be computed for each feature. For instance, one may be interested in the trajectory of two distinct features at the 0.1 and 0.9 quantile levels, respectively, as well as their joint relations, which can be advantageous in many real-world applications (Carlier et al., 2017).

## 4 CONCLUSION

In this work, we have proposed a novel method for conditional generative quantile modelling applied to probabilistic time-series forecasting. While quantile functions offer a unique perspective to generative modelling which may be advantageous in many applications like probabilistic forecasting, extending quantile functions to the multivariate setting is not immediately obvious. Extending existing work on vector quantile regression, we learn a Brenier mapping as our multivariate quantile function where we learn the convex potential by constructing a first-order approximation of the potential w.r.t. our covariate embeddings. Our method is then thoroughly tested on a wide variety of density estimation and time-series forecasting benchmarks and demonstrates strong performance.

ACKNOWLEDGMENTS

We thank the reviewers for their constructive comments, which helped improve our draft. We gratefully acknowledge NSERC, Canada CIFAR AI chairs program, NRC and University of Waterloo for funding support.

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

# A  PRELIMINARY

In this section, we recall some background on generative quantile functions and optimal transport. In the univariate case, given a cumulative distribution function (CDF) $F(x) = \Pr(X \leq x)$ of random variable $X$, the corresponding quantile function $Q$ is defined as: $\forall u \in (0, 1)$,

$$Q(u) = F^{-1}(u) := \inf\{x : F(x) \geq u\}. \tag{6}$$

While the CDF can be straightforwardly extended to the multivariate case, the quantile function, its inverse, is less obvious as the loss of the total ordering on the real line made it impossible to "just invert" $F$. Instead, we can think of the quantile function $Q$ as a mapping that transforms a uniformly random variable $U$ to a sample from the CDF $F$ of interest, be univariate or not. This generative view of the quantile function turns out to be quite fruitful and allows us to learn distributions via quantile regression, through for instance the pinball loss $\ell_u$ at a fixed quantile level $u \in (0, 1)$:

$$\ell_u(q; Y) = u(Y - q)_+ + (u - 1)(Y - q)_-, \tag{7}$$

where $(t)_+ := \max\{t, 0\}$ and $(t)_- := \min\{t, 0\}$. Indeed, for fixed $u$, minimizing the expected loss $\mathbb{E}[\ell_u(q; Y)]$ yields exactly $q = Q(u)$ for the quantile function $Q$ of $Y$ (Koenker & Gilbert, 1978).

**Generative Quantile Networks.** Now, to estimate the entire quantile function $Q$, we can simply replace the fixed quantile level $u$ with a univariate random variable $U \sim \mathrm{Unif}([0, 1])$. We can now minimize the expected loss $\mathbb{E}\ell_U(Q(U); Y)$ with respect to $Q$ to learn the entire quantile function. The univariate quantile loss has been applied successfully recently in probabilistic time-series forecasting. For instance, Gouttes et al. (2021) proposed stacking an implicit quantile network on top of a recurrent neural network (RNN) to model the temporal dynamics of the covariates. By minimizing the quantile loss, exact point-wise quantiles can be inferred when forecasting.

However, when we transition to the multivariate setting where $U$ is uniform over the hypercube $[0, 1]^d$, we can no longer directly minimize the expected loss $\mathbb{E}\ell_U$. A natural alternative is to minimize the $L^p$ norm of the vector quantity in Equation (7). For instance, the autoregressive implicit quantile networks (AIQN) of Ostrovski et al. (2018) amounts to adopting the $L^1$ norm. Wen & Torkkola (2019) proposed to train a Gaussian-copula model jointly on top of a generative multivariate quantile model with the $L^1$-norm quantile loss and account for higher-order correlations in a two-step procedure. Normalizing flows are a popular class of generative models that allow explicit evaluation of the likelihood and the inverse of Jacobian. In particular, Wang et al. (2019) parameterized their quantile function with an increasing triangular map and applied it to novelty detection. However, an extension to conditional quantile estimation was not considered.

**Optimal Transport.** Consider two probability measures $\mu$ and $\nu$ on $\mathcal{X} \subseteq \mathbb{R}^d$ and $\mathcal{Y} \subseteq \mathbb{R}^m$, respectively, and $c : \mathcal{X} \times \mathcal{Y} \rightarrow [0, +\infty]$ a (closed) cost function. The Monge problem refers to finding a transport map $G : \mathcal{X} \rightarrow \mathcal{Y}$ that minimizes the total transport cost:

$$C(\mu, \nu) := \inf_{G:\mathcal{X}\to\mathcal{Y}} \left\{ \int_{\mathcal{X}} c(x, G(x))\mathrm{d}\mu(x) \mid G_{\#}\mu = \nu \right\}, \tag{8}$$

where $G_{\#}\mu$ denotes the push-forward measure by $G$, i.e., if $X \sim \mu$ then $G(X) \sim G_{\#}\mu$. An ubiquitous transport cost is the squared Euclidean distance $c(x, y) = \|x - y\|_2^2$, yielding the (squared) 2-Wasserstein distance, i.e. $\mathcal{W}^2(\mu, \nu)$. A common issue with optimal transport in practice is its computation in high dimensions. Kolouri et al. (2019), among many others, propose to employ random projections and to reduce to the univariate setting with provable convergence guarantees. However, the number of projections sampled from the unit hypersphere necessarily scales with the dimensions of the distribution. Alternatively, Kantorovich's duality offers an objective that can be conveniently approximated by restricting the structure of dual potentials. Formally, the Kantorovich dual problem is given by

$$\sup_{(\varphi,\psi)\in L^1(\mu)\times L^1(\nu)} \int_X \varphi\mathrm{d}\mu(x) + \int_Y \psi\mathrm{d}\nu(y) \tag{9}$$

$$\text{s.t. } \forall x \in \mathcal{X}, \forall y \in \mathcal{Y}, \ \varphi(x) + \psi(y) \leq c(x, y), \tag{10}$$

where $\varphi \in L^1(\mu), \psi \in L^1(\nu)$ can be chosen to be bounded and continuous (Villani, 2008). In fact, the dual should admit a solution, $\frac{1}{2}\|\cdot\|_2^2 - \varphi$ and $\frac{1}{2}\|\cdot\|_2^2 - \psi$ are closed convex functions and are

Fenchel conjugates of each other, provided that $c$ is the squared Euclidean distance. A byproduct of this is that the gradient map $\mathrm{Id} - \partial\varphi$ is an optimal coupling between $\mu$ and $\nu$ (Villani, 2008).

Makkuva et al. (2020) proposed parameterizing $\frac{1}{2}\|\cdot\|_2^2 - \varphi$ as an Input Convex Neural Network (ICNN) (Amos et al., 2017) and optimize the Kantorovich dual for generative modelling tasks. Huang et al. (2021) utilized Brenier's theorem to introduce Convex-Potential Flows (CP-Flows) in which they restrict their flow model with strong convexity. Thus, the flow is the convex gradient that is invertible numerically using a convex solver and they use Hutchinson's trace estimator to estimate the logarithmic Jacobian. Although similar to ours, both works are not conditional and neither are the generative quantile networks since the underlying distribution is not learned through all quantiles jointly.

## B  ESTIMATION OF CPQ

We now turn to the estimation of CPQ. It turns out that directly addressing the maximal correlation functional is a bit challenging. Instead, we resort to its dual problem to learn the Brenier mapping $\nabla g$. Revisiting the primal where we have also introduced a covariate $X$:

$$\sup\{\mathbb{E}(U^\top Y) : U \sim \mu, \mathbb{E}(X|U) = \mathbb{E}(X) = 0\}. \tag{11}$$

Here the mean-independence constraint is added to decorrelate $U$ with the covariate $X$. The dual is derived as the following:

$$\min_{(\varphi,\psi,b)} \int_{[0,1]^d} \varphi \mathrm{d}\mu(U) + \int_{\mathbb{R}^d} \psi \mathrm{d}\nu(Y) \ \text{ s.t. } \ \varphi(U) + \psi(Y) \geq U^\top Y \tag{12}$$

$$\mathbb{E}(X|U) = \mathbb{E}(X) = 0, \tag{13}$$

where $\varphi$ is smooth. We define $\psi$ as the Legendre transformation of $\varphi$,

$$\psi(Y) := \max_{U \in [0,1]^d}\{U^\top Y - \varphi(U)\}, \tag{14}$$

to reduce the problem down to optimizing only $\varphi$ and $b$ as we expect the optimum $\varphi$ to satisfy $\varphi^* = \psi$. By complementary slackness, the inequality in Equation (12) becomes equality for optimal $U$:

$$\varphi(U) + \psi(Y) = U^\top Y. \tag{15}$$

By differentiating Equation (15) w.r.t $U$, we arrive at $Y = \nabla\varphi(U)$. However, this alone is not sufficient for characterizing the existence and convergence of smooth function $\varphi$ to the optimal convex potential gradient that couples $\mu$ and $\nu$. In Appendix C, we provide theoretical justification and necessary conditions for existence, convergence, and duality.

## C  THEORETICAL ANALYSIS

First, we make a remark on the convergence of gradient descent on objective (5).

**Remark.** *Under the Lipschitz-smoothness assumption of $\varphi, b$, and boundedness of $\mathrm{spt}(U)$ and $\mathrm{spt}(X)$, the gradient of objective (5) is bounded. Assuming $N$ is sufficiently large, and random vectors $U$ and $Y$ have finite first moment, then objective (5) is bounded from below (see Proposition 1). Thus, gradient descent converges for objective (5) almost-surely Polyak (1963).*

A necessary assumption is the objective must be bounded from below Polyak (1963). Under some very mild assumptions, we can establish convergence almost surely.

**Proposition 1.** *Assume $N$ to be sufficiently large, random vectors $U$ and $Y$ have finite first moment ($|\mathbb{E}[U]|, |\mathbb{E}[Y]| < \infty$), and $E[f(X)] = 0$. Then, objective (5) is bounded from below.*

Indeed, with further smoothing of objective (5), accelerated gradient methods could be used instead with strong convergence guarantees An et al. (2021). Despite linear parameterizations of $\varphi$ and $b$ by Carlier et al. (2017), the authors established the following three theorems that generalize to smooth $\varphi, b$. These theorems establish the duality between the dual solution and the optimal solution to the correlation maximization primal. First, we are curious if the coupling $(U, Y)$ where $Y := \nabla\varphi_X(U)$ solves the correlation maximization problem in equation 11.

**Theorem 2** (Carlier et al. 2017)**.** *Assume $U \in \mathbb{R}^d$ is random vector with distribution $\mu$, $(X, Y) \in \mathbb{R}^m \times \mathbb{R}^d$ is random vector with joint distribution $\pi$, where $Y \sim \nu$. Furthermore, assume $\mathbb{E}(X|U) = \mathbb{E}(X) = 0$. If there exists smooth function $\varphi : \mathbb{R}^d \to \mathbb{R}$ and smooth function $b : \mathbb{R}^d \to \mathbb{R}^m$ such that $\varphi_X(U) = \varphi(U) + b(U)^\top X$ is convex almost everywhere on the support of $X$ such that $Y = \nabla\varphi(U) + \nabla b(U)^\top X$, then $U$ solves the correlation maximization problem (Eq. (11)).*

For the purpose of our work, $\pi$ will be the joint distribution of $\nu$ and the distribution of the latent embedding space, $\mathrm{Law}(f(X))$. As a consequence of Theorem 2, learning the corresponding Brenier map from $\mu$ to $\nu$ is a surrogate objective for solving the correlation maximization problem. The following theorem suggests the existence of a solution to the dual problem (Eq. (5)) under mild assumptions on the joint distribution $\pi$.

**Theorem 3** (Carlier et al. 2017)**.** *Let $\pi$ be an absolutely continuous probability measure over $\mathbb{R}^m \times \mathbb{R}^d$ with density $g$. Assume the support of $\pi$ is $\bar{\Omega}$ where $\Omega$ is an open bounded convex subset of $\mathbb{R}^m \times \mathbb{R}^d$, and $g$ is bounded on $\Omega$ and bounded away from zero on compact subsets of $\Omega$. Then, the dual problem admits at least one solution.*

So far, we have established the existence of a solution to the dual problem (under mild assumptions) and that if $\varphi_X(U)$ is convex almost everywhere on $\mathrm{spt}(X)$ with respect to $U$, then the corresponding Brenier map coupling solves the correlation maximization problem. The remaining theorem establishes the convexity of the solution to the dual problem (5) with respect to $U$.

**Theorem 4** (Carlier et al. (2017))**.** *Let $U \in \mathbb{R}^d$ be a solution to (Eq. (11)) and let $\varphi : \mathbb{R}^d \to \mathbb{R}, b : \mathbb{R}^d \to \mathbb{R}^m$ be solutions to the corresponding dual problem (Eq. (5)). Let $\varphi_x(u) = \varphi(u) + b(u)^\top x$ $\forall (u, x) \in [0, 1]^d \times \mathrm{support}(\mathrm{Law}(X))$. Then, $\varphi_X(U) = \varphi_X^{**}(U)$ and $U \in \partial\varphi_X^*(Y)$ almost surely.*

By the Fenchel-Moreau theorem (Rockafellar, 1970) and Theorem 4, $\varphi_X(U)$ is convex almost surely. Hence, the transported mass $u \in \mathbb{R}^d$ conditioned on $f(x) \in \mathbb{R}^m$ to $y \in \mathbb{R}^d$ is given by $y = \nabla\varphi_x(u) = \nabla\varphi(u) + \nabla b(u) \odot f(x)$ where $\odot$ denotes the Hadamard product. A property of univariate quantile functions is that they satisfy the non-crossing property. In short, if $u_i \leq u_j$, then $Q(u_i) \leq Q(u_j)$ for any quantile function $Q$. In the multivariate case, the same must hold for each component.

**Proposition 2.** *If $\varphi_X(U) : \mathbb{R}^d \to \mathbb{R}$ is convex with respect to $U \in \mathbb{R}^d$ for each $X$, then each component of the $\nabla\varphi_X(U)$ is monotonically increasing.*

Therefore, convex potential gradients serve as a natural parameterization of quantile networks as additional constraints to enforce the non-crossing property of quantile functions are not needed. Finally, we prove that any conditional quantile function $Q_x(u)$ can be reasonably approximated by the first-order parameterization in Eq. (4).

**Proposition 3.** *For any continuous conditional quantile function $Q_x(u)$ we can find large $n$ and functions $f(x) : \mathbb{R}^m \to \mathbb{R}^n$, $\varphi(u) : \mathbb{R}^d \to \mathbb{R}$ and $b(u) : \mathbb{R}^d \to \mathbb{R}^n$ such that the gradient of $\varphi_x(u) := \varphi(u) + b(u)^\top f(x)$ approximates $Q_x(u)$ uniformly over any compact region of $(u, x)$.*

In our experiments, we find that setting $n = m$ suffices to obtain reasonable results. It would be interesting to explore in future work when and how the approximation can be substantially tightened for certain classes of conditional quantiles and network architectures for parameterizing $b$ and $f$.

## D    PROOFS

**Proposition 1.** *Assume $N$ to be sufficiently large, random vectors $U$ and $Y$ have finite first moment ($|\mathbb{E}[U]|, |\mathbb{E}[Y]| < \infty$), and $E[f(X)] = 0$. Then, objective (5) is bounded from below.*

*Proof.*

$$\frac{1}{N}\sum_{i=1}^N \varphi(U_i) + \frac{1}{N}\sum_{i=1}^N \max_{j \in [N]}\{U_j^\top Y_i - \varphi(U_j) - b(U_j)^\top f(X_i)\} \tag{16}$$

$$\geq \frac{1}{N}\sum_{i=1}^N \varphi(U_i) + \frac{1}{N}\sum_{i=1}^N U_i^\top Y_i - \varphi(U_i) - b(U_i)^\top f(X_i) \tag{17}$$

$$= \frac{1}{N} \sum_{i=1}^{N} \varphi(U_i) + \frac{1}{N} \sum_{i=1}^{N} U_i^\top Y_i - \frac{1}{N} \sum_{i=1}^{N} \varphi(U_i) - \frac{1}{N} \sum_{i=1}^{N} b(U_i)^\top f(X_i) \tag{18}$$

$$= \frac{1}{N} \sum_{i=1}^{N} U_i^\top Y_i - \frac{1}{N} \sum_{i=1}^{N} b(U_i)^\top f(X_i). \tag{19}$$

By the strong law of large numbers,

$$\xrightarrow{a.s.} \mathbb{E}[U^\top Y] - \mathbb{E}[b(U)^\top f(X)]. \tag{20}$$

As $U$ are $Y$ are sampled independently (and so are $U$ and $X$), and $\mathbb{E}[f(X)] = 0$ by assumption,

$$= \mathbb{E}[U]^\top \mathbb{E}[Y] > -\infty \tag{21}$$

as desired. Another perspective which we can view this is we can define sequence $(s_n)$ where $s_n := \frac{1}{n} \sum_{i=1}^{n} U_i^\top Y_i$. Since by SLLN, $s_n \to s := \mathbb{E}[U^\top Y]$, then $(s_n)$ is Cauchy $\implies$ for each (finite) $\epsilon > 0$, there exist $N$ s.t. $\forall n, m \geq N$, $\|s_n - s_m\| < \epsilon \implies |s_n| < \infty$, $\forall n \geq N$. The same can be done for $\frac{1}{n} \sum_{i=1}^{n} b(U_i)^\top f(X_i)$. Thus, combining this result with the boundedness of the gradient, we guarantee convergence of gradient descent Polyak (1963) for $N$ *large enough*. $\square$

**Proposition 2.** *If $\varphi_X(U) : \mathbb{R}^d \to \mathbb{R}$ is convex with respect to $U \in \mathbb{R}^d$ for each $X$, then each component of the $\nabla \varphi_X(U)$ is monotonically increasing.*

*Proof.* Let $H_{\varphi_x}$ denote the Hessian of $\varphi_x$. By convexity of $\varphi_x$, $H_{\varphi_x} \succcurlyeq 0$ so the diagonal entries of $H_{\varphi_x}$ are non-negative. Otherwise if the $i^{th}$ diagonal entry is negative, let $e_i$ be the standard basis at the $i^{th}$ coordinate, then $e_i^\top H_{\varphi_x} e_i < 0$, arriving at a contradiction. Consider the $k^{th}$ component of $\nabla \varphi_x$, $(\nabla \varphi_x)_k = \frac{\partial \varphi_x}{\partial U_k}$, since $H_{\varphi_x} \succcurlyeq 0$ then $\frac{\partial^2 \varphi_x}{\partial U_k^2} = diag(H_{\varphi_x})_k \geq 0$. Thus, $(\nabla \varphi_x)_k$ is monotonically increasing with respect to the $k^{th}$ coordinate. As $k$ is arbitrary, the proposition holds for all $k \in [n]$. $\square$

**Proposition 3.** *For any continuous conditional quantile function $Q_x(u)$ we can find large $n$ and functions $f(x) : \mathbb{R}^m \to \mathbb{R}^n$, $\varphi(u) : \mathbb{R}^d \to \mathbb{R}$ and $b(u) : \mathbb{R}^d \to \mathbb{R}^n$ such that the gradient of $\varphi_x(u) := \varphi(u) + b(u)^\top f(x)$ approximates $Q_x(u)$ uniformly over any compact region of $(u, x)$.*

*Proof.* Fixing $x$, the function $u \mapsto Q_x(u)$ is the gradient of some convex function $q_x(u) =: q(x, u)$. Thus, using results in Chen et al. (2019), we may approximate $q_x(u)$ with a Relu network $g(x, u)$. Using the standard compactness argument, we may approximate $q(x, u)$ over any compact convex region $K$ by $\sum_{i=1}^{n} q_i(x_i, u)$. Now we use the results of Chen et al. (2019) to approximate each $q_i(x_i, u)$ with a convex Relu network $g_i(u)$. Define $f(x) = \mathbf{1}$ and $b(u) = [g_1(u), \dots, g_n(u)]$ we then have $b(u)^\top f(x) = \sum_i g_i(u)$ which approximates $q(x, u)$ uniformly over the compact region $K$. Finally, we note that uniform approximation of a convex function also leads to approximation of its gradient (Rockafellar, 1970).

We note that we can also modify the arguments of Chen et al. (2019) to provide a more direct and possibly tighter proof. $\square$

# E    EXPERIMENTAL DETAILS

## E.1    SAMPLING FROM GAUSSIAN *vs.* UNIFORM

Practically, we found sampling from the unit hypercube sometimes leads to instability in training deep architectures. A similar phenomenon was observed by Wang et al. (2019). Empirically, the model also produced lackluster samples in some applications. Instead, we pre-process the uniform samples using a bijection $\mathbf{\Phi^{-1}} : [0,1]^d \to \mathbb{R}^d$ where $\mathbf{\Phi^{-1}} := (\Phi^{-1}, \dots, \Phi^{-1})$, notably the inverse CDF of a standard normal applied component-wise. Now, the support of the model is unbounded, which is easier to handle computationally as it leads to improved stability in training. Since $\Phi^{-1}$ is monotonically increasing, the solution to the maximal correlation problem is equivalent up to the bijection $\mathbf{\Phi^{-1}}$ due to the rearrangement inequality.

**Remark.** *Alternatively, sampling directly from a standard Gaussian is equivalent as $\Phi^{-1}$ pushes* $\mathrm{Unif}([0,1])$ *forward onto* $\mathrm{Gaussian}(0,1)$. *During inference,* $\Phi$ *can be applied to the Gaussian samples to deduce the corresponding quantile level of each estimate. The support is no longer bounded, however, we perform clipping by sampling from* $\mathrm{Unif}([\epsilon, 1-\epsilon])$ *for small* $\epsilon > 0$ *such that after pushing onto the standard Gaussian, the support is compact in practice. Thus, the theoretical results that follow which assume compactness of* $\mathrm{spt}(U)$ *hold in practice.*

## E.2    COMPETITOR ALGORITHMS

We compare our method against the following deep quantile networks on synthetic data:

- **Spline Quantile Functions (Gasthaus et al., 2019)**. Monotonic splines are used to fit the quantile function and are optimized by minimizing the continuous ranked probability score.
- **Implicit Quantile Networks (Gouttes et al., 2021)**. A neural network is stacked on top of a backbone encoder (i.e. RNN, LSTM) and is optimized by minimizing the (Huber) quantile loss.

Then, we compare our method against the following additional models on multivariate forecasting tasks on real data:

- **DeepAR (Salinas et al., 2019)**. Baseline auto-regressive model for probabilistic forecasting. The model is trained through maximizing the log-likelihood.
- **TimeGAN (Yoon et al., 2019)**. Composed of an auto-encoder trained in parallel with a generator and discriminator. Trained using a reconstruction loss that learns latent embeddings of a sequence, a supervised loss that learns the conditional distribution of the current iterate given past iterates, and an unsupervised loss that learns the joint distribution between the static features and the sequence.
- **TempFlow (Rasul et al., 2021)**. A flow-based model (i.e. RealNVP) is trained jointly with an auto-regressive encoder backbone (i.e. RNN, LSTM). The RealNVP conditioned on the latent embeddings of the sequential covariates predicts the following time-point by maximizing the log-likelihood.

## E.3    PSEUDO-CODE OF DUAL OBJECTIVE

We present the pseudo-code for the correlation maximization dual objective below. Each tensor is assumed to be two-dimensional; the first dimension is the batch axis and the second dimension is the feature axis. Flatten all dimensions except the batch dimension to allow for vector dot product otherwise.

## E.4    MODEL ARCHITECTURE

In our experiments, we test both smooth and input convex neural networks. Empirically, we find that using smooth neural networks, a feasible transport map is learned, however it may not be maximally monotone upon convergence of gradient descent. We can guarantee the mapping is a monotone operator by using input convex neural networks, although expressivity is limited compared to its smooth counterpart.

---

**Algorithm 1:** PyTorch-style code for computing the correlation maximization dual objective.

---

**Input:** $U, \hat{Y}, Y, X$

```
φ, b = Ŷ
Y = Y.permute(1, 0)
X = X.permute(1, 0)
BX = torch.mm(b, X)
loss = torch.mean(φ)
UY = torch.mm(U, Y)
ψ = UY − φ − BX
sup, _ = torch.max(ψ, dim=0)
loss += torch.mean(sup)
return loss
```

---

**Smooth Neural Networks.** $\varphi$ and $b$ can be parameterized using smooth neural networks. The activation of choice must be smooth, so we choose to use CELU activation,

$$\mathrm{CELU}(x) = \max(0, x) + \min(0, \alpha(\exp\left(\tfrac{x}{\alpha}\right) - 1)) \tag{22}$$

for some $\alpha > 0$.

**Input Convex Neural Networks.** We also experiment with input convex neural networks. Here, the activation function is monotonically increasing, and the weights are constrained to be non-negative. Indeed, expressivity may be limited under such constraint.

We implement $f$ as an LSTM unless explicitly stated. We find that setting batch-size $N = 128$ leads to stable convergence on all of our tasks. Full details on implementation details can be found in the following.

Table 2: Summary of quantile network architecture used for each experiment. The number of units in the hidden dimension of the LSTMs is $512$.

| Experiment | Input Dims | # Layers | | # Units | | $f$ |
|---|---|---|---|---|---|---|
| | | $\varphi$ | $b$ | $\varphi$ | $b$ | |
| 2D density estimation | 2 | 3 | 3 | 512 | 512 | Identity |
| Energy | 28 | 3 | 3 | 128 | 128 | 2 Layer LSTM |
| Stocks | 6 | 3 | 3 | 128 | 128 | 2 Layer LSTM |

### E.5 SYNTHETIC 2D EXPERIMENT

The three densities we use are `Eight Gaussians`, `Half Moons`. and `Spirals`. The `Eight Gaussians` dataset does not distinguish points based on label, thus we use this dataset to evaluate the efficacy of our method first as an unconditional density estimator. For conditional density estimation, we turn to `Half Moons` and `Spirals`. Each point in the `Half Moons` dataset is assigned to one of two classes, while each point in the `Spirals` dataset is assigned to one of three classes. As such, $\varphi : \mathbb{R}^2 \to \mathbb{R}^2$ for both datasets, but $b$ maps to $\mathbb{R}^2$ for `Half Moons` and $\mathbb{R}^3$ for `Spirals` with $\mathrm{spt}(b) = \mathbb{R}^2$ for both. Figure 1 shows the generated distributions qualitatively. Here, we set $f(X) = X$. For consistency, we also replace the backbone temporal encoder in SQ-RNN and IQN with the identity mapping. The covariates we pass in for the conditional examples are the one-hot encodings of the corresponding classes. Moreover, for SQ-RNN and IQN, we minimize the CRPS and Huber quantile loss summed across the marginals respectively. For the SQ-RNN implementation, we fit a monotonic spline to each component.

### E.6 MULTIVARIATE PROBABILISTIC FORECASTING

The *UCI Appliances Energy* dataset contains 28 temporal continuous-valued attributes. The *Google Stocks* dataset contains 6 temporal features gathered from 2004 to 2019. For each dataset, we generate time-series samples of 23 contiguous time points with a prediction window of 1. We compare

our method against other competitive probabilistic time-series prediction methods. We use a 2-layer LSTM as the temporal feature extractor, $f$, that yields latent embeddings of the temporal covariates. For consistency, DeepAR, TempFlow, IQN, and CPQ all utilize the same backbone LSTM with the same hyper-parameters. The IQN model is trained by summing across the marginal Huber quantile loss. TimeGAN and RealNVP+RNN cannot be probed to extract the $50^{th}$ and $90^{th}$ quantile levels without empirical estimation which is computationally infeasible.

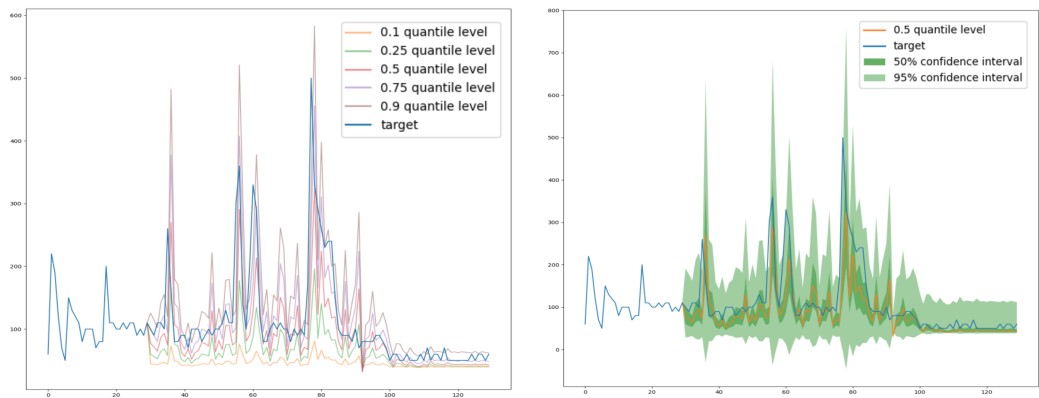

Figure 2: **Estimated test set trajectory of the univariate time-series with prediction window of 100 steps. Left:** The trajectory at varying quantile levels are plotted. **Right:** The corresponding interquartile range and [0.025, 0.975] quantile range computed.

## F  ADDITIONAL EXPERIMENTS AND RESULTS

### F.1  UNIVARIATE TIME-SERIES FORECASTING METRICS

Leveraging a quantile approach to training forecasting models can be quite appealing. Often, predicting a point-wise trajectory is insufficient in modelling risk in fields such as stock analysis for example (Wen & Torkkola, 2019). A naive solution would be to sample the model multiple times to draw an empirical distribution estimation over the trajectory, however, this is computationally expensive and simply does not work for deterministic models. Using a quantile approach, the model only needs to be trained once and any quantile level can be efficiently inferred through one forward pass.

Here, we evaluate if our proposed methodology can be applied to time-series forecasting. Moreover, we demonstrate the main advantage of our quantile approach on forecasting time-series trajectories in a probabilistic flavor. We consider the baseline task of time-series prediction of a univariate response by training and generating trajectories of a single continuous-valued feature from the *UCI Appliances Energy* dataset. In particular, we train our model on samples of 30 contiguous time points to predict the following time point.

Figure 2 presents the trajectory prediction over a 100 time-step interval from the test set. One noticeable difference in our quantile approach to classic forecasting models is that we can precisely and efficiently generate the trajectories associated with a certain quantile risk level (figure 2, top). Naturally, quantile ranges can also be deduced (figure 2, bottom) without the need for empirical estimation.

Quantitatively, we compare our model along with SQ-RNN and IQN (all evaluated at the 0.5 quantile level) against a baseline LSTM trained by minimizing the mean squared error (MSE). According to Table 3, certainly the baseline LSTM performs the best when evaluated using MSE as it was trained by optimizing the MSE. However, comparable performance is observed for our method even when evaluated with MSE. Thus, training using our quantile approach results in a minimal drop in regression performance, while gaining the added advantage of modelling the entire distribution rather than just a point-wise estimate. We note that all the backbone LSTMs in the quantile networks and the baseline LSTM we compare against are consistent.

### F.2  INPUT CONVEX VS. SMOOTH NEURAL NETWORKS

Previous works (Makkuva et al., 2020; Huang et al., 2021) that construct a Brenier map between two distributions parameterize their function as an ICNN. Doing so guarantees that the trained model is convex with respect to the input data. Contrary to these works, our work demonstrates an ICNN

Table 3: **Test set mean squared error for univariate forecasting.** * indicates prediction was computed at the 0.5 quantile level. As expected, LSTM has the lowest MSE as the MSE was directly minimized during training. However, CPQ yields comparable results on MSE and has stronger performance than the other quantile networks.

| LSTM | SQ-RNN* | IQN* | CPQ* |
|---|---|---|---|
| 0.0036 | 0.0136 | 0.0084 | 0.0083 |

Table 4: Performance evaluation on the multivariate time-series datasets `Energy` and `Stocks`. Results are averaged over 5 runs. Lower score is better. We use boldface for the lowest score.

| Dataset | Model | MaxAE | MeanAE | QL50 | QL90 | RMSE | sMAPE |
|---|---|---|---|---|---|---|---|
| Energy | ICNN | 0.863 | 0.063 | 0.030 | 0.018 | 0.100 | 0.246 |
| | Smooth | 0.624 | 0.051 | 0.026 | 0.038 | 0.091 | 0.210 |
| Stocks | ICNN | 0.739 | 0.024 | 0.014 | 0.009 | 0.040 | 0.205 |
| | Smooth | 0.660 | 0.024 | 0.012 | 0.014 | 0.036 | 0.220 |

parameterization is not necessary and that the trained model is still convex with respect to the input as expected from optimal transport theory. Here, we study the effect of restricting the model to be input convex a priori against relaxing this assumption by using smooth neural networks to approximate convex functions. Table 4 displays the results of the two time-series datasets `Energy` and `Stocks` averaged over 5 runs 10 epochs each.

We notice slight improvements of a smooth parameterization compared to the convex parameterization. We suspect this is due to the greater flexibility of smooth neural networks as the weights are not constrained to be non-negative, thus enabling better fitting the optimal convex potential.

### F.3 ABLATIONS

#### F.3.1 EFFECT OF WIDTH AND DEPTH

Optimal transport approaches can suffer in high dimensional settings. Here, we study the effect of high dimensions on our dual objective and suggest a way we empirically found to mitigate the curse of dimensionality. We set the target distribution as a $2048-$dimensional isotropic standard Gaussian. The baseline network is 3-layer $\varphi : \mathbb{R}^{2048} \to \mathbb{R}$ with a 2048-dimensional hidden layer.

We use the Kolmogorov-Smirnov test to compare the closeness of the generated distribution against the target distribution. The statistic is given by,

$$D_n = \sup_x |F_n(x) - F(x)| \tag{23}$$

where $F$ is the cumulative distribution function of the distribution of $X$ (where $X_i$ assumed iid). In essence, this statistic measures the largest variation of the empirical against the target distribution.

In particular, we find that scaling up the depth of the network leads to no improvements in learning the high dimensional target distribution, but scaling up the width leads to improvements. Figure 3 shows the effect of scaling width and scaling depth independently on fitting the high dimensional Gaussian. A theoretical justification is left for future work.

### F.4 ADDITIONAL APPLICATION: IMAGE GENERATION

An interesting area of application for conditional generative quantile models is in the domain of image generation. More specifically, learning the exact quantile levels of certain features could be beneficial in not only image generation, but also image editing and novelty detection. We conduct

a.

b.

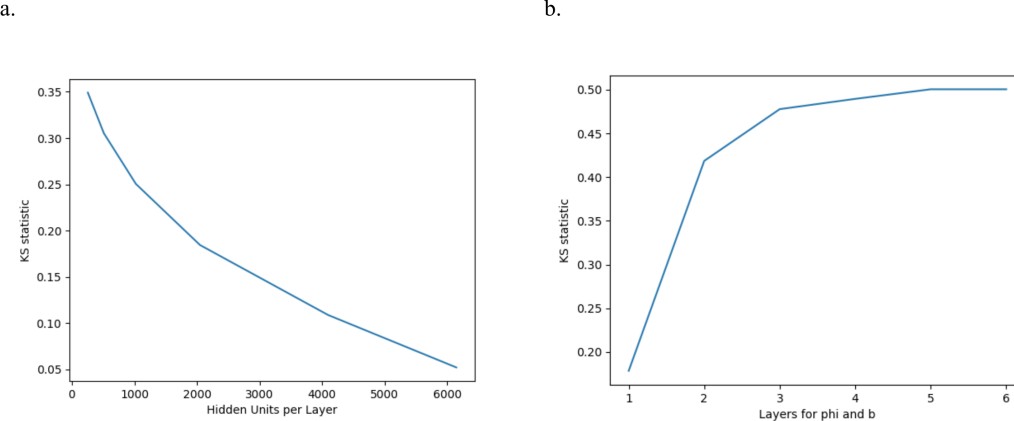

Figure 3: **Kolmogorov-Smirnov Test Ablation on High Dimensional Data.** The target distribution is a 2048-dimensional isotropic standard Gaussian. (a.) The KS statistic with respect to number of hidden units per layer. (b.) The KS statistic with respect to the number of layers. Both $\varphi$ and $b$ are scaled evenly. Lower KS statistic value is better.

some preliminary experiments to demonstrate that our approach can be applied in the image domain. Further applications of replacing our latent embedding encoder $f$ with CNNs are not considered and left for future work.

First, we test our conditional quantile network to generate handwritten digits conditionally. We consider two scenarios; (1) use a variational auto-encoder (Kingma & Welling, 2014) for dimensionality reduction and generate on a latent space (Figure 4.b), and (2) directly generate on the image space (Figure 4.a). The first scenario maps $U \in \mathbb{R}^D$ to $Y \in \mathbb{R}^D$ where $D$ is the dimension of the latent space while the second scenario involves mapping $U \in \mathbb{R}^{784}$ to $Y \in [0,1]^{784}$. The second case is a greater challenge than the first as often we set $D << 784$ and the encoder and decoder can be trained to implicitly filter out noise. However, we show that by directly generating on the image space, high-quality images of the corresponding digit conditioned can be obtained, albeit slightly noisier than the auto-encoder counterpart. Nevertheless, we show that our proposed method can generate high-quality samples in strongly correlated high-dimensional spaces. Moreover, using a dimensionality reduction algorithm such as VAE can improve the visual quality of the generated images by implicitly learning to filter out noise.

Unlike MNIST, CelebA samples have multiple attribute labels rather than a single class label. We consider the model's ability to learn multi-attribute combinations and furthermore generate novel combinations of attributes. The CelebA dataset contains 202,599 RGB images of celebrity portraits aligned and cropped to $64 \times 64\text{px}^2$. Each image corresponds with a $40-$dimensional multi-hot vector that summarizes features of the celebrity portrait such as gender, hair color, and whether the entity is wearing eyeglasses to name a few. A VAE was employed for dimensionality reduction thus our conditional quantile network maps to the latent space. Figure 4.c shows the results of novel attribute combination generation qualitatively.

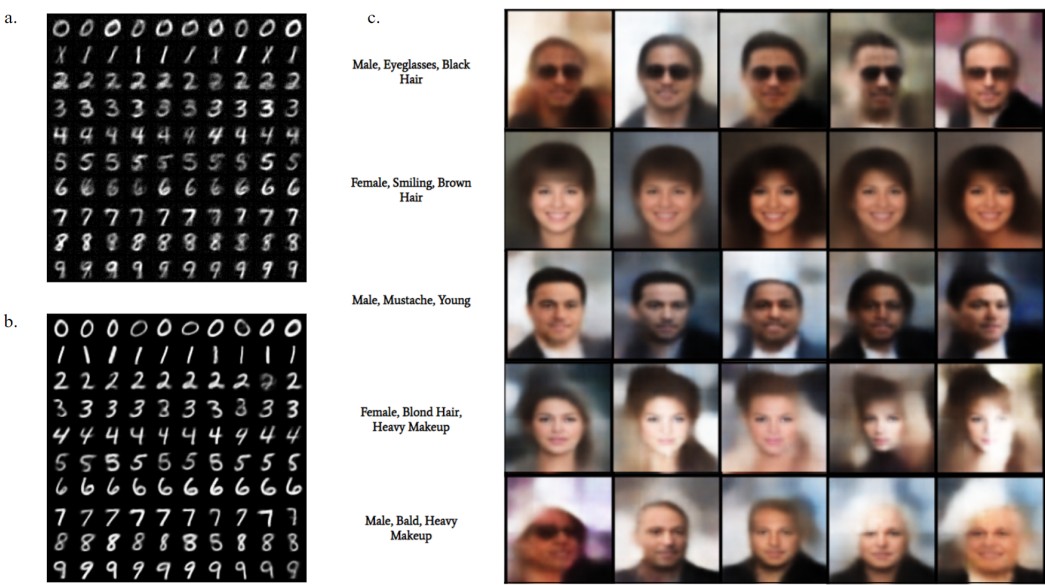

Figure 4: **Qualitative results of generated images.** For the MNIST samples, we conditionally generate the handwritten digits by conditioning on a digit for each row. (a.) Directly learning a mapping onto the image space of MNIST. (b.) Using a VAE, the quantile network maps to the latent space. (c.) Novel combination of attribute generation on CelebA.

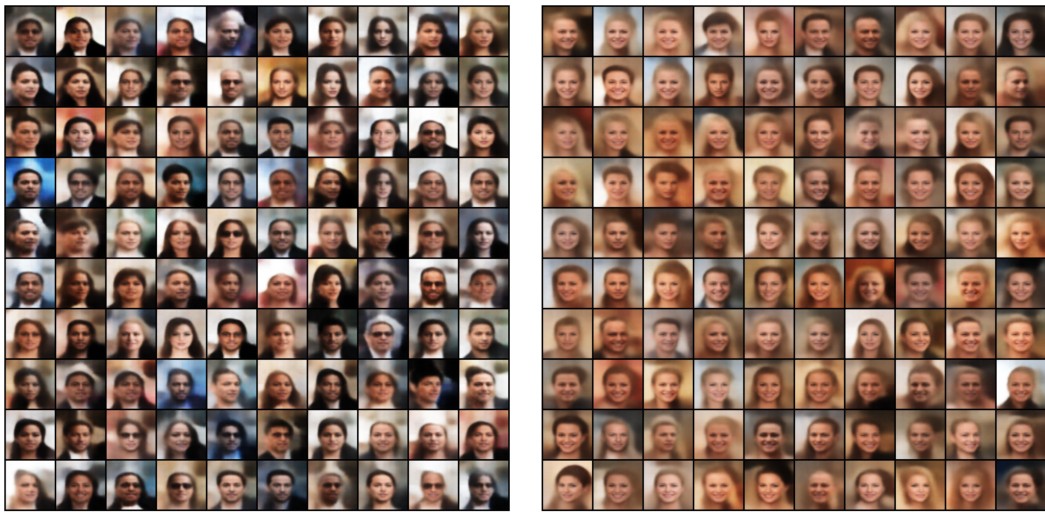

Figure 5: **Left**. Unconditioned. **Right**. Conditioned on Blond, Young, Smiling, Female, Mouth_Slightly_Open.

