# OpenReview forum: "Conditional Generative Quantile Networks via Optimal Transport"
_ICLR.cc/2022/Workshop/DGM4HSD — ICLR 2022 DGM4HSD workshop Poster_

### Official Review · Reviewer_BpyR · 2022-03-24
**Quantile multivariate quantile generative model with optimal transport**

**Rating:** 6
**Confidence:** 2

**Review:**

## Summary

The goal of this paper is to generalize quantile generative modelling using optimal transport to the multivariate case. Quantile regression aims at reproducing the quantile function as an approximation of the data probability distribution. Previous approaches do not perform well in the multivariate case, and this paper builds on early works from Cartier et al. to propose a new method. The latter relies on an optimal transport formulation which aims at maximizing the correlation between a uniform distribution and the target distribution, and for which a dual program can be constructed. The paper includes a series of experiments to compare the performance of their method with previous approaches. In particular, it shows that only their succeed in reproducing some 2d probability density.


## Strengths

- Improvements over other approaches are assessed in several experiments.
- Paper is overall clearly written.


## Limitations

- The theoretical and methodological differences with existing methods could be explained in more details.


## Main review

- sec. 2: At the beginning of the section, the authors say that they "propose a novel method for multivariate density estimation (…)". However, in the rest of the section, it is not clear exactly what are the new contributions compared to earlier paper. They only say that $\varphi$ and $b$ are parametrized by neural networks in their approach. However, the experiments in section 3 do seem to display a clear improvement over existing methods, so it would be useful for the reader to have a clearer picture of the theoretical novelty of the paper which allows for those results.
- sec. 1: Same remark as the previous point, I feel that the introduction could describe in more details what are the new contributions of this paper.


## Clarity

I have found a few minor typos:
- p. 1, last paragraph: "generative quantile modelling Gasthaus et al.": something like "from" is missing before "Gasthaus" (or otherwise parenthesis around the citation).
- p. 2, top of the page: "treating the prediction (…) than the exact point.": I feel that this part of the sentence could be rewritten in a clearer way.
- p. 3, section 3, first paragraph: "max and absolute error, (…) 50th and 90th quantile loss": I would put the plural, "errors", "losses"
- p. 3, section 3, first paragraph: "Description of competitor algorithm is" → "Descriptions of competitor algorithms are"

## Recommendation

I would tend to recommend this paper for the workshop, however, the authors should improve about the motivations as described above.

---

### Decision · Program_Chairs · 2022-03-27

Accept (Poster)